# Changes in Multiple microRNA Levels with Antidepressant Treatment Are Associated with Remission and Interact with Key Pathways: A Comprehensive microRNA Analysis

**DOI:** 10.3390/ijms241512199

**Published:** 2023-07-30

**Authors:** Toshiya Funatsuki, Haruhiko Ogata, Hidetoshi Tahara, Akira Shimamoto, Yoshiteru Takekita, Yosuke Koshikawa, Shinpei Nonen, Koichiro Higasa, Toshihiko Kinoshita, Masaki Kato

**Affiliations:** 1Department of Neuropsychiatry, Kansai Medical University, Osaka 573-1191, Japan; funatsut@takii.kmu.ac.jp (T.F.); ogatahar@takii.kmu.ac.jp (H.O.); takekity@takii.kmu.ac.jp (Y.T.); koshikay@takii.kmu.ac.jp (Y.K.); kinoshit@takii.kmu.ac.jp (T.K.); 2Graduate School of Biomedical & Health Sciences, Hiroshima University, Hiroshima 734-8533, Japan; toshi@hiroshima-u.ac.jp; 3Faculty of Pharmaceutical Sciences, Sanyo-Onoda City University, Sanyo-Onoda 756-0084, Japan; shim@rs.socu.ac.jp; 4Department of Pharmacy, Hyogo Medical University, Nishinomiya 650-8530, Japan; nonen@hyo-med.ac.jp; 5Institute of Biomedical Science, Department of Genome Analysis, Kansai Medical University, Osaka 573-1191, Japan; higasako@hirakata.kmu.ac.jp

**Keywords:** major depressive disorder, microRNA, comprehensive analysis mirtazapine, selective serotonin reuptake inhibitor, antidepressant

## Abstract

Individual treatment outcomes to antidepressants varies widely, yet the determinants to this difference remain elusive. MicroRNA (miRNA) gene expression regulation in major depressive disorder (MDD) has attracted interest as a biomarker. This 4-week randomized controlled trial examined changes in the plasma miRNAs that correlated with the treatment outcomes of mirtazapine (MIR) and selective serotonin reuptake inhibitor (SSRI) monotherapy. Pre- and post- treatment, we comprehensively analyzed the miRNA levels in MDD patients, and identified the gene pathways linked to these miRNAs in 46 patients. Overall, 141 miRNA levels significantly demonstrated correlations with treatment remission after 4 weeks of MIR, with miR-1237-5p showing the most robust and significant correlation after Bonferroni correction. These 141 miRNAs displayed a negative correlation with remission, indicating a decreasing trend. These miRNAs were associated with 15 pathways, including TGF-β and MAPK. Through database searches, the genes targeted by these miRNAs with the identified pathways were compared, and it was found that *MAPK1*, *IGF1*, *IGF1R*, and *BRAF* matched. Alterations in specific miRNAs levels before and after MIR treatment correlated with remission. The miRNAs mentioned in this study have not been previously reported. No other studies have investigated treatment with MIR. The identified miRNAs also correlated with depression-related genes and pathways.

## 1. Introduction

Major depressive disorder (MDD) presents with various symptoms, including depressed mood, low motivation, and feelings of guilt and hopelessness, resulting in severe distress, impairment of personal functioning, social consequences, and increased mortality. The overall annual prevalence of MDD is approximately 6%, the highest for any mental disorder [1]. The lifetime prevalence of MDD is estimated to be approximately 15%, and prospective studies have reported an even higher rate of 38% [2,3]. Even as an actual number, patients with MDD saw increases of 18.4% from 2005 to 2015. Antidepressants are effective to some extent in treating MDD, with these medications growing in societal influence [4]. However, the remission rates, which is the goal of the acute phase, with initial medication are still inadequate, as indicated by several large studies that have shown remission rates of approximately 30%, with the remaining 70% either not responding or only partially responding to treatment [4,5]. The remission rate has also decreased sequentially with the second and subsequent changes in medication [4,6]. Hence, administering the appropriate drug as the first-line treatment is directly related to the wellbeing of the patient [7,8,9,10].

Antidepressants are designed based on the monoamine hypothesis, but the mechanism of their effects is not fully understood; many hypotheses have been proposed and examined. In addition to the involvement of neurotransmitters, multiple factors are thought to be involved in the development of MDD [11,12,13], including gene expression in response to stress load [12,13,14,15], neural plasticity [15,16], immune mechanisms, and inflammatory triggers [12,17,18]. Among them, microRNAs (miRNAs), which regulate gene expression related to depression, have attracted attention as important molecules for treating depression [19,20,21].

miRNAs are endogenous non-coding RNAs, typically 18–25 nucleotides in length [21]. Currently, there are approximately 2000 human miRNAs in the database, many of which are expressed in the central nervous system [19,22,23]. It has been suggested that aberrations in miRNA levels and gene regulatory function in the peripheral blood and central nervous system are associated with MDD and suicide [24,25,26]; however, very few studies have examined the role of miRNAs as biomarkers for treatment response to antidepressant medications [27,28]. To date, three studies have measured the miRNA levels in blood samples before and after treatment and examined their correlation with treatment outcomes for specific antidepressants [29,30,31]. In a study comparing duloxetine with placebo, miR-1456a/b-5, miR-425-3p, and miR-24-3p were associated with treatment response; in another study on escitalopram, desvenlafaxine, and duloxetine, miR-135a-5p, miR-1202, and miR-16-5p were associated with treatment response. Of these three studies, only one study comprehensively analyzed miRNAs [31], and the other two used candidate miRNAs designs [29,30]. All previous studies examined the association of miRNAs with the therapeutic effects of antidepressants, whose primary action is serotonin reuptake inhibition, but not with those of mirtazapine (MIR), whose primary action is the alpha-2-receptor blockade. Additionally, no studies have focused on Asian populations.

The elucidation of miRNAs and their corresponding regulatory genes and pathways that change in relation to treatment outcomes, along with identifying the differences among the drugs with different characteristics, may advance our understanding of the pathophysiology of depression. For this purpose, we aimed to comprehensively investigate the relationship between changes in miRNA levels and treatment outcomes in the plasma before and after drug treatment with two first-line agents for MDD, MIR, and selective serotonin reuptake inhibitors (SSRIs).

## 2. Results

### 2.1. Baseline Analysis

Among the MDD outpatients randomized to MIR or SSRIs in Step I of GUNDAM, the miRNA levels from the start to 4 weeks of treatment by drug administered (ΔmiRNA0-4w) was measured in 46 patients (MIR: 19, SSRIs: 27). There were no significant differences in the clinical and sociodemographic characteristics at the start of treatment between the two groups (Table 1).

### 2.2. miRNA Change and Antidepressant Treatment Response

#### 2.2.1. Mirtazapine

Multiple logistic regression analysis showed a significant association between the primary outcome, ΔmiRNA0-4w, and remission after 4 weeks of treatment for 141 miRNAs in the MIR group (Appendix A). Among the 141 miRNAs, miR-1237-5p showed the most robust association and remained significantly different even after Bonferroni correction for multiple comparisons (odds ratio [OR] = 0.28 (95% confidence interval [CI] = 0.20–0.37), *p* = 1.66 × 10^−2^) (Table 2). Receiver operating characteristic (ROC) curves were generated based on ΔmiR-1237-5p in the MIR group, and the predictive equation for remission after 4 weeks of treatment (AUC = 0.85, 95% confidence interval = 0.72–0.97) (Appendix A). Regarding the secondary outcome, no significantly associated miRNAs were observed in the correlation between the 17-item Hamilton Rating Scale for Depression (HAM-D17) score change from baseline to 4 weeks and ΔmiRNA0-4w.

Given that miRNAs respond with multiple miRNAs rather than single ones, we focused on three miRNAs that showed a strong correlation in the primary analysis ([Top-hit-3s with OR and p value before Bonferroni correction] miR-1237-5p: OR = 0.28, *p* = 3.28 × 10^−5^; miR-4271: OR = 0.61 *p* = 1.03 × 10^−3^, miR-4484: OR = 0.45 *p* = 2.19 × 10^−3^) for further analysis. In Top-hit-3s ΔmiRNA0-4w, both the mean and median changes were decreased in the remission group (Figure 1). Strong (correlation coefficient ≥ 0.8) and positive correlations were observed for the changes from baseline to 4 weeks of treatment for the top three hits (Figure 2). Heat map analysis of the Top-hit-3s ΔmiRNA0-4w in 19 patients treated with MIR was performed as an exploratory approach (Figure 3). Clustering based on changes in the Top-hit-3s levels divided the patients into two groups. Overall, 7 of 19 patients (36.8%) in the MIR group had remissions after 4 weeks of treatment, while 7 of 13 patients (53.8%) in Cluster I (those with poor or negative changes in levels) had remissions. In Cluster II, no case resulted in remission. Interestingly, no patient in Cluster II showed a decrease in miR-4484 level.

#### 2.2.2. SSRIs

No significant associations with ΔmiRNA0-4W were observed for either the primary (remission) or secondary (HAM-D17 change) outcomes. In the SSRI groups, 10 of the 27 patients were in remission and 17 were in non-remission.

### 2.3. miRNA Target Prediction and Pathway Analysis

For the Top-hit-3s associated with the remission of MIR, in silico analysis was performed using DIANA: miRPath v.3 software (University of Thessaly and Information Management Systems Institute (IMSI), Greece). Three pathway databases (micro-T-CDS, Tarbase, and TargetScan) were then used. After conservative analysis based on the probability of a jackknifing test and FDR correction, we identified 15 pathways (Table 3: micro-T-CDS; 11, TagetScan; 5, Tarbase; 0.) that were significantly associated with Top-hit-3s. The FoxO signaling pathway and TGF-β1 signaling pathway showed the most robust correlation with these miRNAs. Preoteoglycan in cancer was identified in two databases. miR-1237-5p was unregistered in microT-CDS but registered in TargetScan. Furthermore, 6 genes were extracted for miR-4271 and 10 genes for miR-4484. The 11 genes on the pathway identified by microT-CDS and the related genes extracted by DIANA TOOLS software “micro-T-CDS” had 4 genes in common (*MAPK1*, *IGF1*, *IGF1R*, and *BRAF*).

## 3. Discussion

The study identified 141 miRNAs that showed pre- and post- treatment changes in levels that correlated with the primary outcome of remission after 4 weeks of treatment in the MIR group. Among them, miR-1237-5p had the strongest correlation and remained significant after correction for multiple comparison testing. Among the primary outcomes, no miRNAs were significantly associated with other treatment outcomes of MIR or overall SSRI outcomes. In the pre- and post-intervention changes in miR-1237-5p, miR-4271, and miR-4484 (i.e., the Top-hit-3s focused upon in this study), a decrease in miRNA expression levels was observed in the MIR group among the remission patients 4 weeks after treatment.

The association of antidepressant-induced changes in miRNA expression levels with treatment outcome has been previously reported [29,31,32,33,34,35,36,37]. miRNAs whose expression levels in the blood is reduced by treatment include miR-34a-5p, miR-221-3p, miR-34c-5p, miR-770-5p, and miR-132 [32,33,35]. miR-1202, miR-124-3p, and miR-134 increase with treatment [29,31,33,36,37]. These studies, with the exception of one [31], had candidate designs and differed from the present study, which comprehensively targeted miRNAs for analysis. Previous studies have specified drugs such as citalopram, escitalopram, paroxetine, duloxetine, desvenlafaxine, and venlafaxine [29,31,32,35,36,38], but no study has used MIR, and many do not specify the drug. In previous studies that were analyzed comprehensively, the drug was restricted to escitalopram; the miRNAs that were downregulated by certain antidepressants and correlated with treatment outcomes were miR-146a-5p, miR-146b-5p, and miR-24-3p, which were observed in the escitalopram response group [31]. This result by Lopez et al. is consistent with that of the present study in that the associated miRNAs were reduced by antidepressants, although the miRNAs involved were different from ours. It is possible that the pharmacological action of antidepressants may reduce certain miRNAs and alter gene regulation, thereby exerting a therapeutic effect. Alternatively, there may be a group of patients in the clinical course of MDD in which certain miRNAs are reduced, and the associated changes in gene regulation may enhance the pharmacological effects of MIR. No previous study has investigated the relationship between pre- and post-treatment expression changes and treatment outcomes for the miR-1237-5p, miR-4271, and miR-4484 identified in this study; however, miR-1237-5p and miR-4271 have been suggested in our previous study as predictors of early treatment at 2 weeks in SSRI-treated patients [39]. In addition, none of the miRNAs mentioned in the previous study, including miR-1202, matched any of the 141 miRNAs that showed significant differences in this study. This is because none of the miRNAs reported in those previous studies met the criteria for analysis in the present study because we only analyzed miRNAs whose samples exceeded 80% of the total samples above the detection sensitivity. Among the miRNAs included in the analysis of this study, miR-1202 has been reported to be significantly upregulated in the remission group by antidepressant treatment, suggesting that it may be a predictor of treatment outcome [36]. However, in this study, there was no significant difference between miR-1202 ΔmiRNA0-4w and the treatment outcome. Furthermore, the blood expression level of miR-1202 in patients with MDD before treatment was inversely correlated with treatment response to antidepressants [29,36]. In our previous study [39], the miR-1202 expression level before treatment in the SSRI groups was inversely correlated with response after 2 weeks of treatment, which is consistent with previous studies [36]. One of the major differences between this study and previous ones is that the latter did not focus on MIR. This study is novel because no previous study has investigated miRNAs whose changes in expression levels correlate with treatment outcomes in MIR. Unlike SSRIs, MIR does not inhibit serotonin reuptake but promotes the release of serotonin and norepinephrine by specifically blocking alpha-2 adrenergic receptors. It also decreases appetite and insomnia symptoms through the blockade of serotonin histamine H1 receptors and adrenergic alpha-1 receptors. These differences seem to elucidate that the discovery of miR-1237-5p, miR-4271, and miR-4484, remained unmentioned in previous studies (scrutinizing the correlation between pre- and post-treatment expression alterations and treatment outcomes), and were first identified in the current study.

The Top-hit-3s ΔmiRNA0-4w showed a correlation coefficient of more than 0.8, indicating a strong association. Based on this correlation, we combined the results from microT-CDS, a database in which miR-1237-5p is not yet registered, with those from TargetScan, a database in which it is. Among the 15 pathways identified, the FoxO and TGF-β signaling pathways showed the strongest association with miRNAs reduced by mirtazapine, with 20 and 12 genes associated with each pathway, respectively. Proteoglycans in cancer was the only pathway identified in both microT-CDS and TargetScan. The MAPK signaling pathway, which previous studies have reported to be associated with miRNAs associated with depression and antidepressant treatment, was also identified in this study. Regarding the molecules comprising the FoxO signaling pathway, which was associated with miR-4271 and miR-4484, FoxO transcription factors are involved in cell apoptosis and the maintenance of homeostasis (Appendix A). Particularly, the inactivation of the PI3K/Akt/FoxO3a circuit may play an important role in the pathophysiology of MDD [40,41]. Brain-derived neurotrophic factor and protein kinase B are the top mediators of FoxO transcription factors, and they are involved in neurogenesis and synapse formation [42]. In addition, although not consistent with the drugs used in this study, antidepressants altering FoxO phosphorylation and the possibility that antidepressant administration may be related to treatment response via the higher expression of mediators was considered in this study [43].

Regarding the molecules that comprise the TGF-β signaling pathway associated with miR-4271 and miR-4484, TGF-β1 is an anti-inflammatory cytokine known to be protective against neurodegeneration and is involved in synaptogenesis plasticity (Appendix A). The plasma levels of TGF-β1 are significantly decreased in MDD and are correlated with its severity, with more effective antidepressant treatment responsibility in the group with higher plasma levels of TGF-β1 [44]. Furthermore, antidepressants may increase TGF-β1 by inducing the release of TGF-β1 or by affecting acquired immune mechanisms, but there are no reports about MIR regarding this [45,46]. TGF-β1 production in patients with MDD also decreases after treatment, suggesting that it is involved in maintaining homeostasis in response to MDD [47]. In this study, the TGF-β signaling pathway was associated with miRNAs whose expression levels were significantly reduced in the MIR-treated remission group. The possibility that the miRNAs identified in this study may have a mechanism through which to increase the production of TGF-β1 by decreasing its expression levels was considered. In our previous study [39], miRNAs that correlated with early response in SSRI-treated patients were extracted and reported to be involved in the TGF-beta signaling pathway. This suggests that TGF-β1 may be involved in MDD pathogenesis.

Regarding the molecules constituting the MAPK signaling pathway, which is affected by miR1237-5p, the strongest correlation in this study, *MAPK1*, is involved in neuroplasticity and inflammation, and its downregulation reduces depressive-like behaviors in a mouse model of depression. In particular, the miR-129-5p expression level has also been found to be associated with that of *MAPK1* [48]. In addition, *MAPK1* has been suggested to correlate with the treatment resistance of MDD with antidepressants [49,50]. It remains unclear how the miRNAs extracted in this study increase or decrease *MAPK1*; however, if they act in a pro-inflammatory manner against *MAPK1*, this would be consistent with the results of previous studies linking the involvement in inflammation and neuroplasticity to treatment outcomes in depression [13,18]. In a previous study, miR-425-3p and miR-24-3p, which were not included in the analysis in this study, showed decreased expression after antidepressant treatment in the treatment response group, suggesting that they were associated with *MAPK1* [31]. The fact that the Top-hit-3s in this study were correlated with *MAPK1* suggests that the pathways targeted by the miRNAs may be similar, even if the miRNAs are different from those in previous studies.

The Proteoglycan in cancer pathways was the only one affected by all three miRNAs. Proteoglycans are complexes of sugars and proteins, such as hyaluronic acid and chondroitin sulfate. They are found in extracellular matrices and cell surfaces throughout the body, including organs, the central nervous system, and skin [51,52,53]. Proteoglycans have been associated with immune responses in malignant tumors, and Proteoglycan in cancer is a pathway that occurs within this association [54]. On the other hand, they are also a major component of the extracellular matrix and cell surface in the central nervous system, suggesting that they may regulate neuroplasticity and neuroaxonal migration [51,52,53]. Hyaluronic acid is thought to be involved in maintaining neuroplasticity and cognitive learning functions within the central nervous system by forming macromolecular assemblies with lectican CS proteoglycans [52,55]. We cannot refer to specific proteoglycans in this study; however, Proteoglycan in cancer is the only pathway identified in the two databases, and the association of proteoglycan with neuroplasticity and immune response is likely relevant to the pathogenesis and therapeutic response of MDD. Two miRNAs correlated with Proteoglycan in cancer (miR-4271, miR-4486) showed a trend toward decreased expression levels in the MIR-treated remission group, suggesting that these miRNAs might have suppressed neuroplasticity and anti-inflammatory effects through increased expression levels.

Among the genes on the above pathways, *MAPK1*, *IGF1*, *IGF1R*, and *BRAF* were strongly associated with each of the miRNAs by in silico analysis (conducted using the DIANA TOOLS software “micro-T-CDS”). The relationship between *MAPK1* and MDD has been discussed previously. IGF1 (insulin-like growth factor-1) is a polypeptide similar in molecular structure to insulin, and—in addition to its weak insulin-like effect in the body—it is involved in the regulation of DNA synthesis [56], including cell growth, proliferation, and programmed cell death [57]. IGF1R is a specific receptor for IGF1. An association between *IGF-1* and affective disorders has been hypothesized based on its physiological mechanisms [58,59]; however, there are currently no significant reports in humans. In a rat model of depression, *IGF-1* exerted its antidepressant effects through the PI3K/Akt/Fox3 pathway [60]. There are very few reports on *BRAF*, suggesting that it is associated with MDD; however, future research needs to be closely monitored.

We are confident that the results of this study are novel and meaningful. However, there were some general limitations to the study design. First, the results of this study, although based on rigorous statistical analysis including the Bonferroni correction, are limited because of the small cohort size. Therefore, the possibility of an increased risk of overfitting cannot be denied. At this significance level, our sample had the power of 0.80 to detect a large effect size (OR = 0.29 for miR-1237-5p) in the MIR sample (m = 19) between remission and non-remission; therefore, smaller effects could have been missed [61]. This means that a validation with a large cohort of patients with MDD is needed to assess the association and specificity more accurately in the extracted miRNAs, pathways, and genes. Second, the placebo and/or no-treatment groups were not included in the analysis. This study was based on a randomized controlled trial design, which can reduce known and unknown biases; however, the use of placebo and no-treatment groups serves as a valid strategy through which to distinguish between response to antidepressants, placebo, and to the spontaneous improvement of symptoms. Third, the results of this study were observed for a period of 4 weeks; hence, they cannot be applied to the overall treatment of depression, which is a chronic condition. Yet, in the acute treatment of depression, the early treatment results of up to 4 weeks are known to significantly correlate with the subsequent 6–14 weeks of continued treatment [62,63,64]. The results of the present 4-week treatment can thus be appropriately interpreted as applicable to the acute-phase treatment. Another limitation of this study was that the validation of each miRNA expression using qPCR was not performed. However, qPCR and the non-amplified microarray used in this study (3D-Gene) have different detection methods, resulting in large variations in the detectable miRNAs [65,66]. In recent publications, miRNA levels have been evaluated only via 3D-Gene, as described above, without qPCR validation [67,68,69]. In addition, we used a strict threshold expression level for the detection of each miRNA as mentioned in the Methods section. The strength of this study is that we comprehensively examined the miRNA expression levels before and after treatment, as well as examined the relationship between miRNA and the prediction of treatment outcomes from multiple perspectives. In addition, the use of MIR and the fact that the study was limited to Asians are novel.

## 4. Materials and Methods

### 4.1. Study Design and Participants

This study is part of Step I of the Genotype Utility Needed for Depression Antidepressant Medication (GUNDAM) trial [5]. The GUNDAM study has been outlined as a two-step, open-label, randomized, flexible-dose, 8-week trial. Of these, Step I (the first 4 weeks) evaluated treatment response to MIR and SSRI, which have different pharmacological actions in untreated patients with depression. The goal was to individualize the first-line drugs for MDD treatment and combination therapy as a second option based on biological and clinical factors. The study was registered with the University Hospital Medical Information (UMIN, number 000006417). The study patients were Japanese outpatients aged 20–75 years. They met the diagnostic criteria for MDD according to the Diagnostic and Statistical Manual of Mental Disorders, Fourth Edition, Structured Clinical Interview for Axis I Disorders and had a score of 14 or higher on HAM-D17. They had not taken psychotropic medications for at least 14 days prior to participation in this study. The diagnoses were made by two independent senior psychiatrists and confirmed by a third psychiatrist. Patients were excluded if they had a clinically significant medical condition requiring treatment, a major psychiatric disorder other than major depression, a history of psychoactive substance addiction or dependence syndrome within the past 6 months, pregnancy, or a history of electroconvulsive therapy within the past 6 months. Participants were randomized equally into the MIR or SSRI (paroxetine or sertraline) groups at inception. The initial doses of MIR, paroxetine, and sertraline were 15, 10, and 25 mg/day, respectively. Once tolerability was confirmed, each drug was increased to 30, 20, and 50 mg/day, respectively, within 2 weeks and to 45, 40, and 100 mg/day, respectively, within 4 weeks. HAM-D17 was administered every other week from the beginning to the end of the study as a clinical evaluation scale for MDD, and a score of ≤7 was defined as remission.

### 4.2. Microarray Analysis of miRNA Expression

Pretreatment and 4-week post-treatment human plasma samples obtained from patients with MDD were used. Total RNA was extracted from 300 µL of serum samples using the 3D-Gene RNA extraction reagent supplied in the liquid sample kit (Toray Industries, Kanagawa, Japan). Comprehensive miRNA expression analysis was performed using the 3D-Gene miRNA Labeling kit and 3D-Gene Human miRNA Oligo Chip (Toray Industries), which are designed to detect the 2555 miRNA sequences registered in the released miRBase 20 databases (http://www.mirbase.org/) (accessed on 2 March 2022). The miRNA assays that detected less than 80% of the analytes were considered unreliable. Of the 2555 miRNAs assessed, the data obtained for 684 were considered sufficiently reliable and used for subsequent statistical analysis, having initially undergone quantile normalization. Signals below the mean background signal value of ± 2 SD, or those considered abnormal in the scan images and were inappropriate for data, were removed as missing areas.

### 4.3. Statistical Analysis

Among the randomized patients assigned to the treatment group at baseline and at the 4-week HAMD17 assessment, those with reliable miRNA data, as described above, were included in the analysis. All participants were divided into remission and non-remission groups. The association between remission after 4 weeks of treatment and changes in miRNA levels from the start to 4 weeks of treatment by drug administered (ΔmiRNA0-4W) was used as the primary outcome. In the secondary outcome, changes in HAM-D17 score at 4 weeks was used instead of the remission at 4 weeks. Factors such as age, gender, illness duration, and pretreatment severity have been reported to influence the treatment outcomes for depression, and multicollinearity was excluded before using them as dependent variables in this analysis [70,71,72,73,74,75,76,77]. These dependent variables were used to generate ROC curves at ΔmiR-1237-5p, and remission at 4 weeks of treatment (Appendix A). Logistic regression models were used for the binary outcomes and multiple regression models for continuous variables. Results with alpha levels of <0.05 were considered significant. The models were controlled for age, sex, duration of current depressive episode, the HAM-D17 total score before treatment, and ΔmiRNA0-4W. Bonferroni correction was used for the proportion of miRNAs that were likely to be identified as significant by chance via multiple testing, with a significance level of *p* < 7.31 × 10^−5^, from the 684 analyzed miRNAs. The correlation coefficient between each ΔmiRNA0-4W that was associated with the treatment outcome was calculated using Pearson’s correlation. For other continuous data between the two treatments, we performed an analysis of variance (ANOVA) or Wilcoxon rank sum test if the assumption of normality was violated. For binary outcomes, the differences in proportions between the two groups were analyzed using the chi-square and Fisher’s exact tests, as required. To generate heat maps, we performed hierarchical cluster analysis with complete linkage to identify patient clusters, based on ΔmiRNA0-4W. The Euclidean distance was used to measure the dissimilarity between each pair of observations, under a hierarchical clustering approach to identify clusters of patients. Statistical analyses for the primary and secondary outcomes were performed using the R Statistics Package v.3.51 (R Foundation for Statistical Computing, http://www.R-project.org) (accessed on 2 March 2022). To validate our miRNA target prediction and pathway analyses, we performed in silico analysis using DIANA: miRPath (v.3). This analysis software facilitates the identification of common miRNA targets, using DAVID EASE scores with conservative stations based on jackknifing test probability and the Benjamini–Hochberg False Discovery Rate (FDR) correction that is derived from DIANA microT-CDS (v5.0), TarBase v7.0 (University of Thessaly and Information Management Systems Institute (IMSI), Greece), and the target scan database. We used the top three miRNAs for pathway analysis, even if more than three were significant, based on previous experience that has indicated that the use of an excessive number of miRNAs can lead to difficulties in interpreting the results [78]. The DIANA TOOLS software “micro-T-CDS” was used to search for genes associated with Top-hit-3s and MDD.

## 5. Conclusions

This study demonstrated that ΔmiRNA0-4w is associated with remission after 4 weeks of treatment with MIR in 141 miRNAs. miR-1237-5p exhibited the strongest correlation yet has remained unmentioned in previous studies. Top-hit-3 miRNAs, miR-1237-5p, miR-4271, and miR-4484, having a high correlation coefficient with each other, were downregulated in the MIR remission group. In silico pathway analysis identified 15 pathways that are involved in MDD, including FoxO, TGF-β, and MAPK1, and their association with Top-hit-3 miRNAs. Among the genes extracted from these pathways, MAPK1, IGF1, IGF1R, and BRAF were identified to have a robust association with Top-hit-3s by in silico analysis. Our findings elucidate a correlation between the attenuation of miRNA levels and symptom remission after 4 weeks of treatment with MIR, indicating that the association of miRNAs with MDD outcomes, particularly with specific antidepressants, may contribute to understanding the pathophysiology of MDD and antidepressant mechanisms of action. Further studies, including basic research on the miRNAs isolated in this study, will be needed to substantiate this supposition. We aim to persist in our research and further authenticate the results of this study through the analysis of a more substantial sample population.

## Figures and Tables

**Figure 1 ijms-24-12199-f001:**
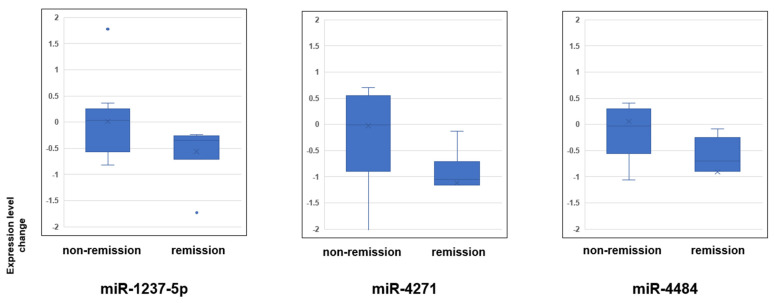
Box and whisker plot showing the changes in the Top-hit-3s levels in the MIR group, the changes were categorized separately for the remission and non-remission groups after 4 weeks. The miRNA expression analysis was performed with a 3D-Gene miRNA Labeling kit and 3D-Gene Human miRNA Oligo Chip (Toray Industries, Inc., Kanagawa, Japan). The horizontal lines in each box indicate the quartile, and the whiskers indicate the maximum and minimum values. Comparison between treatment remission and treatment non-remission groups (*t*-test, *p* < 0.05).

**Figure 2 ijms-24-12199-f002:**
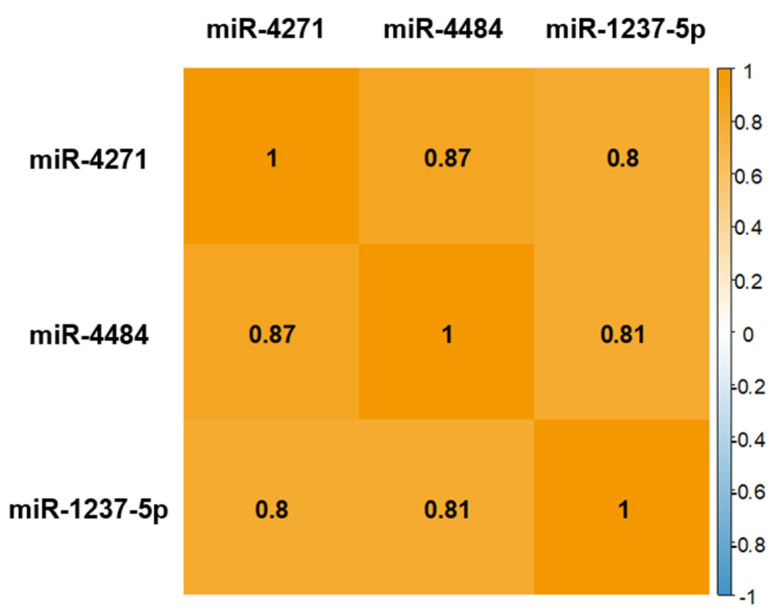
A correlation matrix among the Top-hit-3 miRNAs significantly correlated with treatment remission in patients with major depressive disorder who were treated with mirtazapine for 4 weeks. The numbers indicate the correlation coefficients.

**Figure 3 ijms-24-12199-f003:**
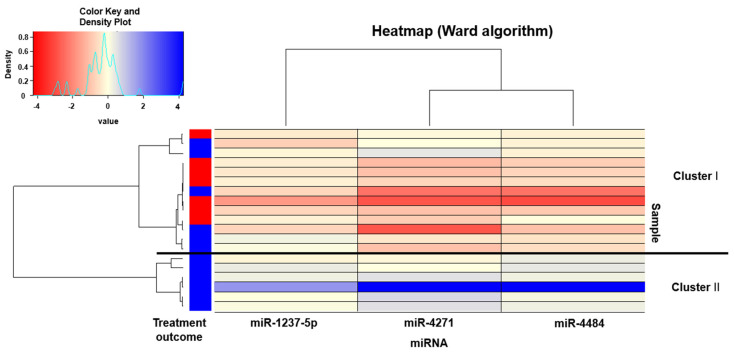
In a group of MDD patients treated with mirtazapine for 4 weeks, hierarchical clustering was performed to identify the Top-hit-3 miRNAs significantly correlated with treatment remission, and this was based on the changes in miRNA levels before and after treatment. Treatment outcome—red: remission; blue: non-remission. In the boxes, red indicates the decreased expression levels and blue indicates the increased expression levels. The color key and density plot are illustrated in the upper left.

**Table 1 ijms-24-12199-t001:** Demographic and clinical characteristics of patients with depression who participated in the study.

	Total (*n* = 46)		Mirtazapine (*n* = 19)		SSRIs (*n* = 27)		* *p*
	%		%		%		
Sex (women)	37.0%		36.8%		37.0%		n.s.
First episode	63.0%		57.9%		66.7%		n.s.
Physical comorbidity	34.8%		31.6%		37.0%		n.s.
Family history	35.6%		33.3%		37.0%		n.s.
Smoking	0%		0%		0%		n.s.
Drinking	31.8%		33.3%		30.8%		n.s.
Occupational status: Employed	82.6%		78.9%		85.2%		n.s.
	Mean	SD	Mean	SD	Mean	SD	
Age	46.3	14.8	48.5	14.7	44.8	14.9	n.s.
Age of onset	42.9	15.2	43.8	15.7	42.3	15.1	n.s.
Duration of current MDD episode (months)	7.0	12.6	3.9	7.1	8.8	14.7	n.s.
HAM-D 17 items total score	21.3	5.4	22.2	6.3	20.4	4.6	n.s.

Abbreviations—SSRIs, selective serotonin reuptake inhibitors; MDD, major depressive disorder; and HAM-D, Hamilton depression rating scale. * *p* < 0.05.

**Table 2 ijms-24-12199-t002:** Results of the multiple logistic regression analysis for the primary outcome (Top-hit-3 miRNAs).

miRNA	*p*-Value	Adjusted *p*-Value (Bonferroni Correction)	Odds Ratio
miR-1237-5p	3.28 × 10^−5^	1.66 × 10^−2^	0.28
miR-4271	1.03 × 10^−3^	n.s.	0.61
miR-4484	2.19 × 10^−3^	n.s.	0.45

**Table 3 ijms-24-12199-t003:** Results of the in silico analysis for the Top-hit-3s associated with remission of MIR. For each of the 15 pathways extracted, the database used *p*-value, total number of genes, and number of genes per miRNA, which are presented.

KEGG Pathway	Database	*p*-Value	Total Gene Number	miRNA (Number of Related Genes)
FoxO signaling pathway	microT-CDS	1.31 × 10^−4^	20	miR-4271 (8), miR4484 (13)
TGF-beta signaling pathway	microT-CDS	1.31 × 10^−4^	12	miR-4271 (8), miR4484 (4)
Lysine degradation	microT-CDS	1.63 × 10^−4^	7	miR-4271 (2), miR4484 (5)
Renal cell carcinoma	microT-CDS	3.74 × 10^−4^	13	miR-4271 (9), miR4484 (4)
Arrhythmogenic right ventricular cardiomyopathy (ARVC)	TargetScan	9.71 × 10^−4^	12	miR-1237-5p (7), miR-4271 (5)
Signaling pathways regulating the pluripotency of stem cells	microT-CDS	9.96 × 10^−4^	19	miR-4271 (9), miR4484 (11)
Hippo signaling pathway	microT-CDS	3.56 × 10^−3^	12	miR-4271 (2), miR4484 (10)
Glioma	microT-CDS	3.56 × 10^−3^	10	miR-4271 (6), miR4484 (4)
Proteoglycans in cancer	microT-CDS	8.57 × 10^−3^	18	miR-4271 (11), miR4484 (7)
	TargetScan	1.18 × 10^−2^	16	miR-1237-5p (9), miR-4271 (7)
ErbB signaling pathway	microT-CDS	1.37 × 10^−2^	13	miR-4271 (10), miR4484 (3)
Chronic myeloid leukemia	microT-CDS	1.37 × 10^−2^	12	miR-4271 (9), miR4484 (3)
Oxytocin signaling pathway	TargetScan	1.76 × 10^−2^	20	miR-1237-5p (10), miR-4271 (10)
Endometrial cancer	microT-CDS	2.25 × 10^−2^	9	miR-4271 (4), miR4484 (6)
MAPK signaling pathway	TargetScan	3.50 × 10^−2^	26	miR-1237-5p (10), miR-4271 (17)
Huntington’s disease	TargetScan	3.50 × 10^−2^	12	miR-1237-5p (6), miR-4271 (6)

## Data Availability

All the data generated or analyzed during this study are included in this article and its online Appendix A. Further inquiries can be directed to the corresponding authors.

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
