# Peer review of "Changes in Multiple microRNA Levels with Antidepressant Treatment Are Associated with Remission and Interact with Key Pathways: A Comprehensive microRNA Analysis"

_ijms, 2023, doi:10.3390/ijms241512199_

Round 1

Reviewer 1 Report

In this manuscript, Toshiya Funatsuki et al. analysed the changes in plasma microRNAs correlated with outcome of depression treatment. The authors used microarray analysis to identify 3 microRNAs (1237-5p, 4271 and 4484) downregulated in the mirtazapine remission group. Next, using an in silico analysis they propose a mechanism correlating the attenuation of microRNA levels and remission after treatment. The authors correctly identified the study design limitations. The results presented are interesting but there are significant issues with methodology and reporting of results which I have outlined below. I would recommend a major revision.

Major comments:

RESULTS

1.     Multiple logistic regression analysis typically requires a large sample size.  A general guideline is that a minimum of 10 cases for each independent variable are needed. Given the mirtazapine group size (19) only SINGLE logistic regression is appropriate! If this was what the authors actually did then consider adjustment!

2.     Given what the authors found on detecting responders in mirtazapine group, the lack of ROC analysis is surprising. Please consider.

METHODS

3.     There are no data on validation of microarray results (ideally qPCR).

4.     I urge the authors to consistently review the entire statistical analysis.

Minor comments:

1.     Line 23: genes are typically TARGETED by miRNAs

2.     Line 27: extracted miRNAs sounds confusing, identified or found is more appropriate!

3.     Line 204: “the extraction of miRNAs” is confusing. Please be more specifically.

4. Line 323 ends abruptly, probably the authors should continue: … and a score of <7 was defined as remission.

Reviewer 2 Report

Study period shall be mentioned in the abstract as well as in the M&M section.

In the introduction, rationale of the study needs improvement.

Cohort size of the patients is small and the duration of the study is 4 weeks. Authors shall explain the validity and applicability of the obtained results in the manuscript.

Round 2

Reviewer 1 Report

The revised version of Toshiya Funatsuki et al. manuscript addressed all my major comments. I recommend this paper for publication.